# Pervasive allele-specific regulation on RNA decay in hybrid mice

Wei Sun[1,2,*], Qingsong Gao[2,*], Bernhard Schaefke[1], Yuhui Hu[1], Wei Chen[1,3]

**Cellular RNA abundance is determined by both RNA transcription and decay. Therefore, change in RNA abundance, which can drive phenotypic diversity between different species, could arise from genetic variants affecting either process. However, previous studies in the evolution of RNA expression have been largely focused on transcription. Here, to globally investigate the effects of *cis*-regulatory divergence on RNA decay in mammals for the first time, we quantified allele-specific differences in RNA decay rates (ASD) in an F1 hybrid mouse. Out of 8,815 genes with sufficient data, we identified 621 genes exhibiting significant *cis*-divergence. Systematic analysis of these genes revealed that the genetic variants affecting microRNA binding and RNA secondary structures contribute to the observed divergences. Finally, we demonstrated that although the divergences in RNA abundance were predominantly determined by allelic differences in RNA transcription, most genes with significant ASD did not exhibit significant difference in RNA abundance. For these genes, the apparently compensatory effect between the allelic differences in RNA transcription and ASD suggests that changes in RNA decay could serve as important means to stabilize RNA abundances during mammalian evolution.**

## Introduction

Eukaryotic gene expression is regulated at multiple steps, and the balance between two opposing biological processes, RNA transcription and its decay, determines the cellular abundance of RNA transcripts (Garneau et al, 2007; Dolken et al, 2008; Schwanhausser et al, 2011; Rabani et al, 2011, 2014). Although to date most studies on RNA expression regulation were focused solely on transcription, recent works have clearly demonstrated the important role of RNA decay (Raghavan et al, 2002; Hao & Baltimore, 2009; Schwanhausser et al, 2011; Rabani et al, 2011, 2014). Often, in response to an extrinsic or intrinsic stimulus, the RNA decay rate can change rapidly to adjust the RNA levels with or without transcriptional change (Raghavan et al, 2002; Hao & Baltimore, 2009). Such regulation is mediated by the interaction between *cis*-regulatory elements residing within the RNA transcripts and diffusible *trans*-acting factors, including RNA-binding proteins (RBPs) and regulatory RNAs such as microRNAs. During the past decades, a number of *cis*-elements have been identified (Caput et al, 1986; Shaw & Kamen, 1986; Xia et al, 1996; Bartel, 2004; Mendell et al, 2004; Vlasova et al, 2008; Ivanov & Anderson, 2013), and importantly, genetic variants affecting these *cis*-elements often alter the RNA decay rate and can result in pathological phenotypes (Rodningen et al, 1998; Xia et al, 1998; Wang et al, 2008; Puimege et al, 2015; Khabar, 2017; Patel et al, 2017).

Changes in RNA expression constitute one of the major forces driving both phenotypic diversity among individuals within the same species (Albert & Kruglyak, 2015) and evolutionary divergence between different species (Necsulea & Kaessmann, 2014). Such changes could arise from genetic variants affecting either transcription or decay. However, because most previous studies analyzed only the effects of genetic variants on steady-state RNA expression levels, they could not distinguish the effects on transcription from those on decay and thus could not elucidate the underlying regulatory mechanisms. To address this, the Gilad and Pritchard labs analyzed the individual-specific mRNA decay rates of more than 16,000 genes in 70 Yoruba HapMap lymphoblastoid cell lines and identified 31 genes with significant *cis*-RNA decay quantitative trait loci (rdQTLs) at a false discovery rate (FDR) of 15% (Pai et al, 2012). To increase their detection power, they then focused on single-nucleotide polymorphisms (SNPs) already identified as steady-state expression QTLs (eQTLs) (Pai et al, 2012). Out of 1,257 eQTLs, 195 were also significantly associated with variations in mRNA decay rates. Interestingly, among the joint QTLs, whereas in 55% cases, the alleles with higher steady-state level decay slower, the remaining 45% showed the opposite pattern of allelic bias between the steady-state expression and RNA decay.

A more direct approach to estimate the *cis*-regulatory effect on RNA degradation is to compare the allele-specific decay rates of RNA transcripts in an F1 hybrid (Dori-Bachash et al, 2011, 2012;

[1]Department of Biology, Southern University of Science and Technology, Shenzhen, China [2]Laboratory for Functional and Medical Genomics, Berlin Institute for Medical Systems Biology, Max-Delbrück-Centrum für Molekulare Medizin, Berlin, Germany [3]Medi-X Institute, SUSTech Academy for Advanced Interdisciplinary Studies, Southern University of Science and Technology, Shenzhen, China

Correspondence: chenw@sustc.edu.cn
*Wei Sun and Qingsong Gao contributed equally to this work.

Andrie et al, 2014). Those allelic transcripts are subject to the same *trans*-regulatory environment, so that observed allelic differences should reflect the impact of *cis*-regulatory divergence. Recently, several studies have investigated allele-specific differences in mRNA decay rates (ASD) for F1 hybrids between different genetically diverse yeast strains (Dori-Bachash et al, 2011, 2012; Andrie et al, 2014). Strikingly, in all these F1 hybrid studies in yeast, for more than 80% of the genes with significant allelic biases in mRNA decay (ASD), their allele-specific mRNA decay and allele-specific RNA expression biased toward opposite alleles, suggesting pervasive compensatory effects between the evolutions of RNA transcription and RNA decay. Such occurrence (>80%) of compensatory effects observed in yeast is much higher than that (45%) observed in the aforementioned human rdQTLs study (Pai et al, 2012). Compared with unicellular organisms such as yeast, more complex gene regulation would be required in multicellular organisms with various organs and cell types. Therefore, such different observation may reflect different evolutionary modes of gene expressing between yeast and mammals. However, alternatively, it can also be due to the different designs of these studies (QTLs versus F1 hybrid). To finally tackle this question, a direct genome-wide profiling of allele-specific RNA decay patterns in multicellular species, such as mammals, would be necessary.

Here, to globally investigate the effects of *cis*-regulatory divergence on RNA decay in mammals, we quantified ASD in an F1 hybrid between two inbred mouse strains, *Mus musculus* C57BL/6J (BL6) and *Mus spretus* SPRET/EiJ mouse strain (SPRET). These two mouse strains diverged ~1.5 million years ago, resulting in ~35.4 million SNPs and ~4.5 million insertions and deletions (indels) between their genomes (Dejager et al, 2009; Keane et al, 2011). Such a high sequence divergence allowed us to unambiguously determine the allelic origin for a large fraction of sequencing reads, thereby enabling accurate measurement of ASD for thousands of genes. In total, out of 8,815 genes with sufficient data for accurate quantification of ASD, we identified 621 genes (7.0%) exhibiting significant *cis*-divergence. Compared with genes without allelic bias, those with ASD divergence contained higher densities of sequence variants. Systematic analysis of sequence features of the genes with biased allelic decay revealed that miRNA-binding sites within 3′ untranslated regions (UTRs) and the local RNA secondary structure in both coding regions and 3′ UTRs could affect RNA decay. Finally, via investigating the role of ASD in the allele-specific RNA abundances (ASA), we demonstrated that on one hand, the observed ASA divergences were predominantly determined by the allelic differences in RNA transcription (AST) and on the other hand, most (>80%) of the genes with significant ASD did not exhibit significant ASA, indicating the pervasive compensatory effects between AST and ASD also existing in mammalian evolution and suggesting that changes in RNA decay rates could serve as important means to stabilize RNA abundances during evolution.

# Results

## Pervasive allelic divergence on RNA decay rates in an F1 hybrid mouse

To investigate the allelic divergence of RNA decay rates in a mammalian system, we measured the ASD in a fibroblast cell line derived from an F1 hybrid mouse between the BL6 and SPRET strains. As shown in Fig 1, we monitored the changes of the allelic RNA abundances following transcriptional arrest using actinomycin D. More specifically, paired-end sequencing was performed on poly-A RNA samples isolated from two biological replicates of F1 fibroblast cells collected at 0, 0.5, and 1.5 h subsequent to transcriptional arrest. On average, each sample yielded 130.1 million read pairs (Table S1). Fig S1 shows the good reproducibility between the two replicates for all the three time points. The high density of sequence variants between the genomes of BL6 and SPRET enabled unambiguous assignment of allelic origin for an average of 62.5 million read pairs in each sample (Table S1; see the Materials and Methods section for details).

To estimate the allele-specific RNA decay rate in a quantitative manner, we used the reads with unambiguous allelic origin. More specifically, we used only the reads that were mapped on SNP loci within genic regions. After filtering out the SNP loci with potential allelic read mapping bias due to the incomplete SNP annotation in paralogous genes or pseudogenes, 8,815 genes containing at least five SNPs supported with sufficient allelic reads were retained (Fig S2; see the Materials and Methods section for details).

To identify the genes with significant ASD, we combined a previously published logistic model and a bootstrapping strategy (Andrie et al, 2014; Muzzey et al, 2014). In brief, we assumed an exponential decay model for each allele. For each time point (0, 0.5, and 1.5 h after transcriptional arrest), the read counts derived from one allele given the total were modeled by a binomial distribution. After logit transformation, the parameters could be directly estimated using a linear logistic model in which the regression coefficient for time variable represents the mRNA decay rate difference $\Delta\lambda = \lambda_1 - \lambda_2$ between the two alleles (see the Materials and Methods section for details). To assess the significance of ASD, we then applied a bootstrapping strategy to estimate the confidence of estimated $\Delta\lambda$. Specifically, for each gene consisting of a list of at least five SNP loci, we generated 5,000 new lists, each consisting of the same number of SNP loci that were chosen at random with replacement from the original list. For each of the 5,000 random lists, $\Delta\lambda$ was estimated using the same logistic model, and altogether yielded a bootstrap distribution, which was then summarized with a mean and a standard deviation. The larger the bootstrap mean deviates from zero, the larger the decay rate diverges between the two alleles. In contrast, lower bootstrap standard deviation gives higher confidence in the estimation of $\Delta\lambda$. According to the bootstrap mean and standard deviation, the statistical significance of ASD was then determined for each gene. After applying a threshold of Benjamini–Hochberg–adjusted *P*-value < 0.05 and $|\Delta\lambda| > 0.06$ in both replicates (FDR = 4.18%; Fig S3), we identified 621 (7.0%) genes exhibiting significant ASD (Fig 2A). Fig 2B shows two representative examples with significant ASD, biased toward the BL6 and the SPRET allele, respectively.

To assess the accuracy in quantifying ASD based on short Illumina reads, we randomly selected 25 genes for independent experimental validation. Using the PacBio RS system, we deep-sequenced the RT–PCR products amplified from samples collected at 0 and 1.5 h, using primers targeted at the regions with no sequence variants between the two alleles (see the Materials and Methods section). The longer read length allowed the assignment of the PacBio reads to the parental alleles without any ambiguity.

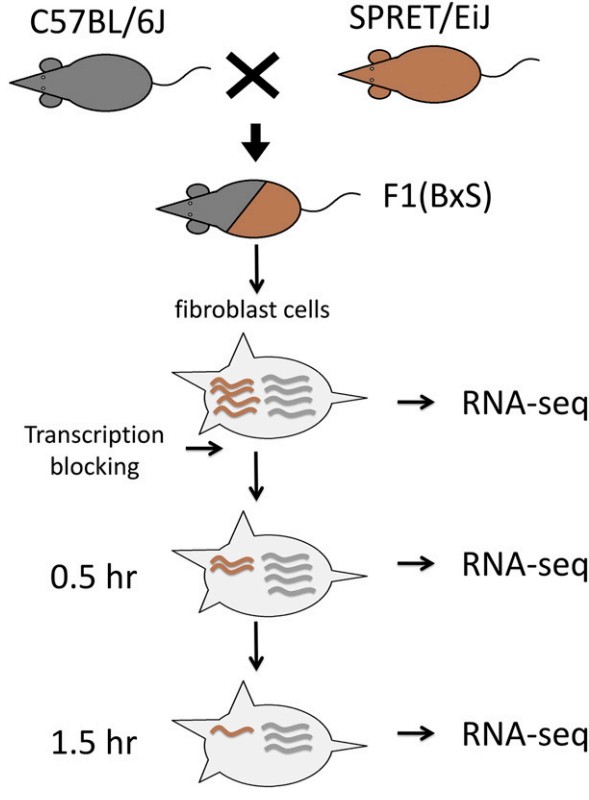

**Figure 1. Overview of experimental design.**
Fibroblast cells were isolated and cultured from the adult F1 hybrid mice between C57BL/6J and SPRET/EiJ. Two replicates of RNAs collected at three different time points following transcriptional arrest were sequenced.

Allelic ratios of the read counts were then compared between the two time points. As shown in Fig 2C, the allelic decay rates estimated in this way were significantly correlated with those determined using the Illumina approach ($r_{Pearson}$ = 0.93, $P-$value $< 2.0 \times 10^{-11}$).

## Genomic features that correlate with ASD divergences

The ASD divergences observed in F1 cells should reflect the effect of the sequence variants influencing *cis*-regulatory elements within the RNA transcripts. To study the potential *cis*-features accounting for the observed allelic biases, we first calculated the frequencies of sequence variants for the genes with or without significant ASD. As shown in Figs 3A and S4, the genes with significant ASD (621) exhibit significantly higher density of sequence variants than the genes without significant ASD (1,319 control genes); $P-$value $< 2.2 \times 10^{-16}$, two-sided Kolmogorov–Smirnov test (see the Materials and Methods section for details).

Next, we sought to identify the potential *cis*-elements accounting for such ASD divergences. Given the well-known importance of miRNA in regulating RNA stability, we first focused on the variants affecting miRNA target sites (Bartel, 2004). For this purpose, we predicted for both alleles the target sites of the miRNAs expressed in the F1 fibroblasts using *TargetScan* (Friedman et al, 2009) (see the Materials and Methods section). Then we compared the number of miRNA-binding sites between the two alleles for the genes with

significant ASD and 621 control genes with similar variant density, but without allelic divergence in decay rates, separately (Fig S5; see the Materials and Methods section for selection of these control genes). For the top 50 highly expressed miRNAs, Fig 3B shows that the difference in the number of their binding sites between the stable (slow-decaying) allele and the unstable (fast-decaying) allele centered symmetrically around zero for the control group (the stable allele was randomly selected here). In contrast, for the ASD genes, the distribution is not symmetric: the unstable alleles tend to possess more miRNA target sites than the stable alleles, demonstrating the contribution of allelic differences in miRNA regulation to the observed ASD. The same trend holds true for the top 100 highly expressed miRNAs (Fig S6) and also holds true when predicting miRNA target sites using a different algorithm, *miRanda* (Enright et al, 2003) (Fig S7). It is known that miRNAs confer the regulation mainly through binding to the targeting sites at 3' UTR regions. Therefore, we further separated the genes into coding regions, 5' UTR and 3' UTR, predicted the miRNA-binding sites, and repeated the same allelic comparison for the three regions separately. Interestingly, the significant contribution of allelic difference in miRNA-binding sites could only be observed for 3' UTR regions, consistent with the canonical model of miRNA regulation (Fig S8).

RNA secondary structure has been reported to regulate RNA decay (Skripkin et al, 1990; Hamilton et al, 1999; Park & Maquat, 2013; Spitale et al, 2015). To check if the sequence variants affecting RNA secondary structures contribute to the observed ASD, we calculated the minimal free energy (MFE) of RNA segments (20-nt flanking each SNP) along the whole transcript for the two alleles separately using *RNAfold* (Lorenz et al, 2011), and then compared the allelic differences between ASD genes and control genes (see the Materials and Methods section). As shown in Fig 3C, compared with the control genes, the ASD genes indeed exhibited larger allelic differences in MFE values (|ΔMFE|). The trends remain regardless of the length of RNA fragments used for MFE calculation (Fig S9) and also holds true when calculating MFE using a different algorithm, *RNAstructure* (Bellaousov et al, 2013) (Fig S10). We again separated the genes into coding regions, 5' UTR and 3' UTR, and repeated the analysis for the three regions separately. As shown in Fig S11, interestingly, larger allelic MFE differences in ASD genes could be observed in both the CDS regions and 3' UTR regions, but not in the 5' UTR regions.

In previous studies, a number of additional sequence motifs have also been reported to affect RNA stability. One of such *cis*-elements is the well-known AU-rich elements (AREs) (Shaw & Kamen, 1986). It has been demonstrated that depending on the RBPs recruited, AREs could either stabilize or destabilize the host RNA transcripts (Garcia-Maurino et al, 2017). To investigate whether AREs also accounted for the ASD observed in this study, we calculated the ARE difference between the two alleles using the program *AREScore* (Spasic et al, 2012) (see the Materials and Methods section). However, as shown in Fig S12, no significant difference in allelic ARE divergence was observed between the control and ASD gene groups. Codon usage has recently been shown to play an important role in regulating mRNA stability (Bazzini et al, 2016; Mishima & Tmari, 2016). Here, to investigate whether codon usage differences between the two alleles contributed to the ASD observed in this study, we calculated the codon usage biases of the two alleles using codon adaptation index

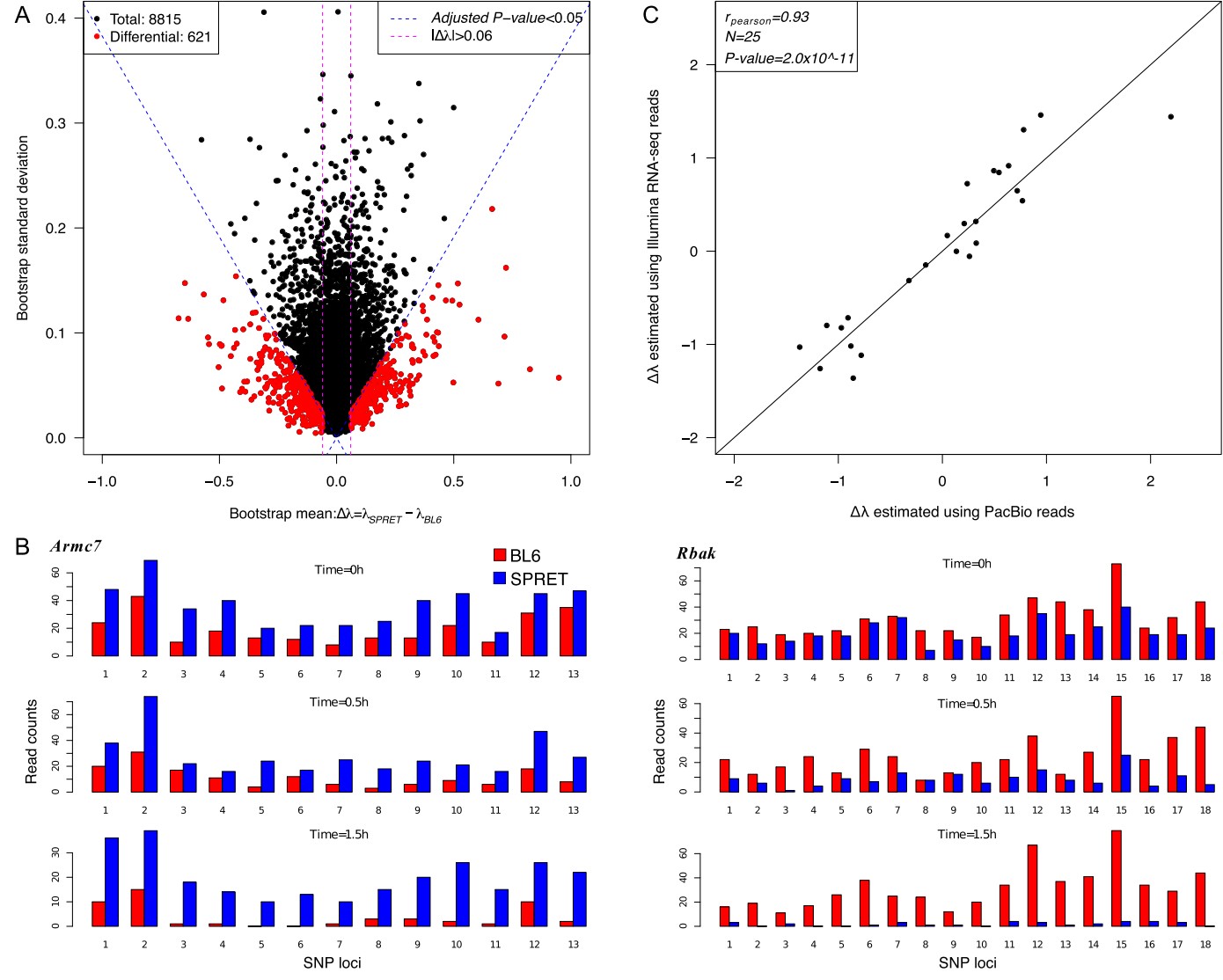

**Figure 2. Identification of genes with significant ASD.**
**(A)** Scatterplot showing the bootstrap means (x-axis) and standard deviations (y-axis) of estimated ASD. Dashed blue lines indicate the Benjamini–Hochberg–adjusted P-value of 0.05 and dashed purple lines indicate a minimum decay rate difference of 0.06. Out of 8,815 genes (black), 621 (red) exhibited significant ASD. **(B)** Bar plots showing the number of sequencing reads assigned to BL6 (red) or SPRET (blue) alleles (y-axis) at different SNP loci (x-axis) of three time points (0, 0.5, and 1.5 h). BL6 and SPRET allele degraded faster in *Armc7* and *Rbak* genes, respectively. **(C)** Scatterplot comparing allelic decay rate difference (Δλ) estimated based on Illumina sequencing data (y-axis) to that based on PacBio sequencing (x-axis) for the 25 randomly selected genes. Δλ estimated based on the two technologies was significantly correlated ($r_{Pearson} = 0.93$, $P − value < 2.0 × 10^{-11}$).

(Sharp & Li, 1987), but we did not observe any significant correlation between the allelic difference in codon usage and the observed ASD (Fig S13; see the Materials and Methods section).

**The role of ASD in the allelic difference of RNA abundances**

In the F1 hybrids, the allele-specific bias in RNA abundance (ASA) results from the balance between AST and ASD. Previous studies in yeast using similar hybrid systems have demonstrated that the allelic biases in the two processes often possess opposite effects on the RNA abundance and some of the evolutionary changes in RNA decay are mechanistically coupled with those in RNA transcription

(Dori-Bachash et al, 2011). Considering the higher complexity of gene regulation, here based on our dataset, we sought to address in a mammalian system whether and how the two processes, ASD and AST, coordinated with each other. For this purpose, we first investigated the relative contribution of the ASD to the ASA, the latter being estimated based on our poly-A RNA sequencing data collected at 0 h (steady state, before transcription arresting). Using the same bootstrapping strategy on log2 fold change of allelic expression at the same FDR threshold (adjusted P-value < 0.05, allelic divergence greater than twofold, FDR = 4.76%), out of the 8,815 genes for which we could confidently measure ASD, we identified 1,241 genes exhibiting ASA divergence (Figs 4 and S14).

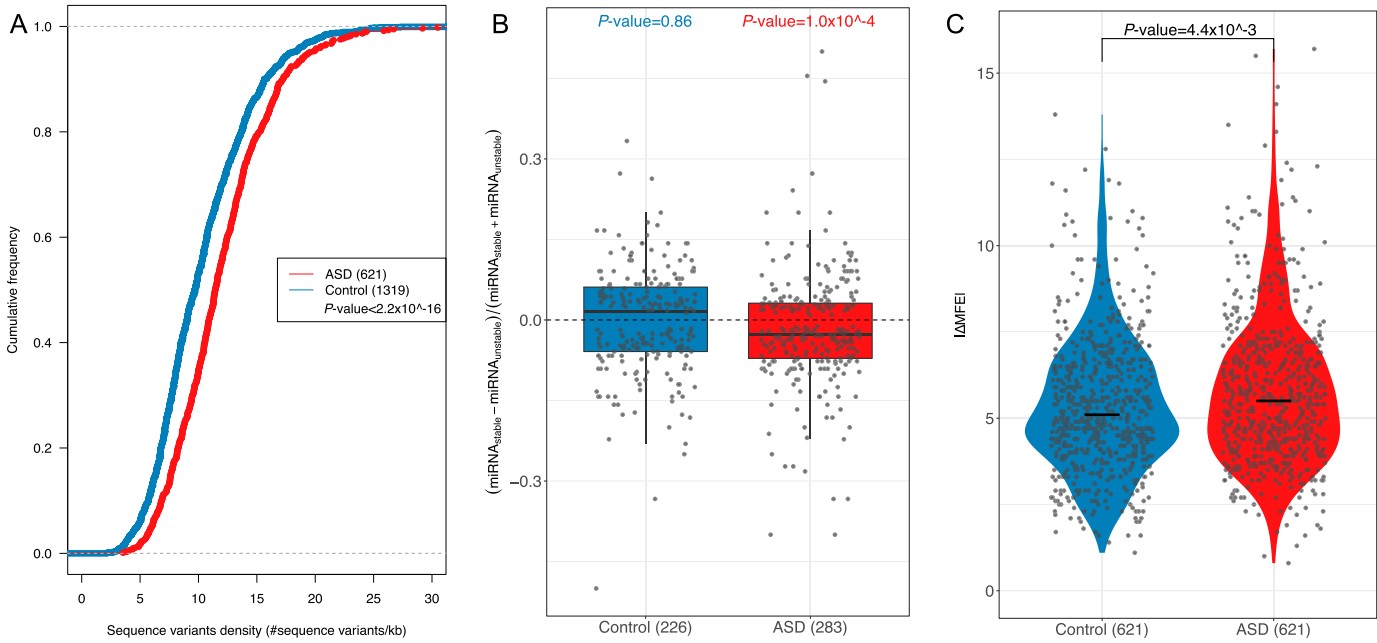

**Figure 3.  Sequence features that were correlated with ASD.**
**(A)** The cumulative distribution function (CDF) of SNP density (number of SNPs per kb) for genes with significant ASD (red) and without (control genes, blue). Compared with the control genes, the genes with significant ASD showed significantly higher SNP density ($P$-value $< 2.2 \times 10^{-16}$, two-sided Kolmogorov–Smirnov test). **(B)** Box plots and scatterplots showing the distribution of miRNA-binding site number difference between the stable and unstable alleles for genes with significant ASD and controls. For controls, the difference centered around zero ($P$-value = 0.86, two-sided Mann–Whitney $U$ test), whereas in ASD genes, unstable alleles tend to possess more miRNA target sites than the stable alleles ($P$-value = $1.0 \times 10^{-4}$, two-sided Mann–Whitney $U$ test). Only the genes with ≥10 miRNA-binding sites combining the two alleles together and ≥1 different sites between the two alleles were used. **(C)** Violin plots and scatterplots comparing the distribution of the absolute MFE difference ($|\Delta MFE|$) between ASD genes and controls. The horizontal lines indicate the median. Compared with controls, ASD genes exhibited larger allelic differences ($P$-value = $4.4 \times 10^{-3}$ two-sided Mann–Whitney $U$ test).

To study the role of ASD in ASA, we then compared the genes with significant ASA to those with ASD. On one hand, most of the 1,241 genes with significant ASA (1,136 genes, 91.5%) did not exhibit significant ASD (Fig 4), suggesting that *cis*-divergence in RNA decay did not contribute much to the observed ASA. Instead, the ASA should largely result from the significant allelic biases in RNA transcription. On the other hand, among the 621 ASD genes, most (516 genes, 83.1%) did not exhibit significant ASA (Fig 4). For these genes, allelic bias in RNA transcription and that in RNA decay have opposite effects on the RNA abundances. To avoid the effect of arbitrary thresholds, we used different combinations of FDRs for ASA and ASD. As shown in Fig S15, the trend is consistently observed at different cutoffs.

## Discussion

The cellular abundance of RNA transcripts is determined by the balance between RNA transcription and decay. Therefore, change in RNA expression could arise from genetic variants affecting either/both of the processes. In spite of this, most of the previous studies in the evolution of RNA expression have been largely focused only on transcription. To globally investigate *cis*-divergence of RNA decay in mammals, we conducted a first genome–wide ASD profiling in a hybrid mouse system, the F1 cross between the BL6 and SPRET inbred mouse strains. Among all the mouse strains with

high-quality genome assembly, SPRET has the largest number of sequence variants relative to BL6, which provides a large number of potential regulatory variants between the two strains (Keane et al, 2011). In total, out of 8,815 genes with sufficient data for accurate quantification of allelic difference in RNA decay rates, we identified 621 genes (7.0%) exhibiting significant *cis*-divergence, indicating widespread *cis*-divergences in RNA decay.

To distinguish the effects of transcription from those of decay on the changes of RNA abundance, the Tirosh lab investigated the evolutionary divergence in mRNA decay between closely related yeast species and their F1 hybrid (Dori-Bachash et al, 2011). Interestingly, they found that nearly 80% of the genes with differences in both mRNA degradation and steady-state levels and decay and transcription had opposing effects. In a later study by the Akey Lab, comparing the ASD in an F1 hybrid of two genetically diverse yeast strains, a similar phenomenon was observed. These studies suggest that in yeast, RNA transcription and decay are evolved in an opposite manner, indicating strong stabilizing selection for steady-state RNA expression levels. Compared with simple organisms such as *Saccharomyces* yeasts, much higher complexity is often required in the regulation of gene expression in multicellular species with various organs and cell types. Thus, it is an intriguing question whether the observed evolutionary patterns of RNA transcription and RNA decay in yeast also hold true for higher organisms, such as mammals. In a population study of the human interindividual variations in RNA decay, although a significant proportion (45%) of

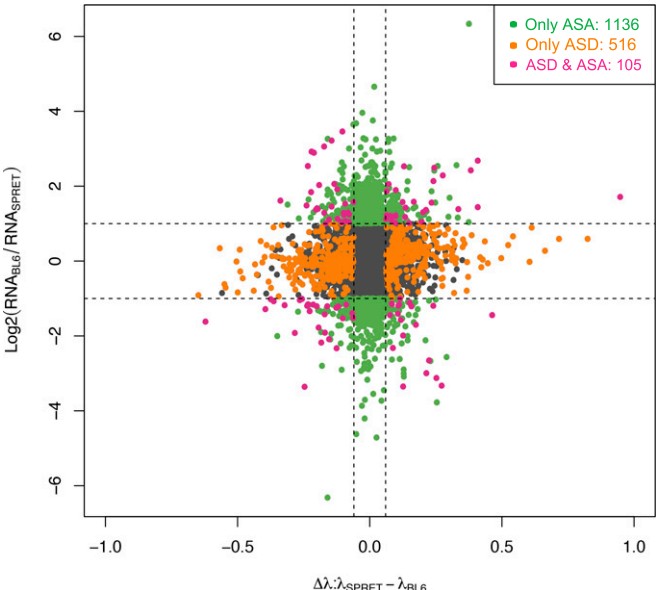

**Figure 4. The role of ASD in the allelic difference of RNA abundances.**
Scatterplot comparing each gene's allele-specific expression (log2-transformed fold change at *y*-axis) and decay (Δλ at *x*-axis). Dashed gray lines indicate twofold change for gene expression and 0.06 for decay rate difference, respectively (FDR < 0.05). Genes with significant allelic bias at only RNA abundance level, only decay level, and both levels were depicted in green, orange, and purple, respectively.

rdQTLs exhibited opposite effects to those of RNA transcription (inferred from steady-state mRNA expression levels), this proportion is much smaller than that identified from yeast studies (>80%). There are at least four different possible (not necessarily mutually exclusive) scenarios explaining the different observations between yeast and human: 1) Mechanical coupling for opposing effects in transcription and decay is not as prevalent in mammals as in *Saccharomyces* yeasts. 2) Pai et al (2012) sought to identify the rdQTLs within the set of significant eQTLs. In this case, if the effect of one rdQTL balanced the effects of other variants (such as QTLs on RNA transcription) on the mRNA expression level of the target gene, resulting in no significant variation among the population, then there would be no eQTL identified for this target gene. Consequently, with this study design, compensatory rdQTLs would be largely ignored and the total amount of rdQTLs as well as the proportion of rdQTLs with opposing effects to transcription remained largely underestimated. 3) Gene expression regulation in yeasts and mammals evolved along different trajectories, with stronger stabilizing selection in *Saccharomyces* than mammals. Such scenario would be consistent with the vastly greater effective population size of yeasts relative to that of mammals. 4) The divergence time and reproductive isolation between two yeast species or strains is much larger than between variants stemming from the same human population. Therefore, no evidence for compensatory evolution would be expected in the latter case, assuming random mating with regard to the QTLs studied and the absence of population substructure.

In this study, comparing the genes with significant ASD to those with significant ASA, we observed that the majority (1,136 out of 1,241, 91.5%) of the genes exhibiting ASA showed no significant ASD,

indicating that allelic difference in transcription should be the predominant contributor to the observed ASA. This observation is in agreement with the previous human QTL study of RNA decay, in which the authors found that most (84.5%) of the identified eQTLs (expressed RNA abundance QTLs) were not rdQTLs (Pai et al, 2012). Taken together, it is likely, in mammals, that most of the divergence on the cellular RNA abundances results from the changes of RNA transcription. Interestingly and more importantly, we observed that 83.1% of the genes with significant ASD did not show allelic biases in RNA abundances, suggesting *cis*-divergences on RNA transcription and decay in these genes have opposite effects on RNA abundance. This indicates that pervasive opposing effects between transcription and decay observed in yeast also exist in mammals. The second scenario discussed above most likely explained the observation in the previous human QTL study.

The opposite *cis*-divergent effect could result from two possible scenarios. First, a mechanistic coupling between RNA transcription and decay, where the same *cis* change simultaneously leads to an increase (decrease) in transcription and an increase (decrease) in RNA decay. Second, to stabilize the RNA abundance, a change causing increased (decreased) transcription (or decay) is followed by an independent change causing increased (decreased) decay (or transcription). By comparing the parental differences and the allelic differences for both RNA transcription and RNA decay in the yeast hybrid system, Dori-Bachash et al (2011) distinguished the *cis-/trans*-origins of the divergences in RNA transcription and RNA decay. Interestingly, for those genes with opposite effects on RNA transcription and decay, the divergences of RNA transcription and decay often originated either both from *cis* or both from *trans*, suggesting that these opposite divergences might result from the same genetic variants, thus mechanistically coupled. Further analyses indeed suggested that the changes in some *trans*-factors (such as Rpb4/7 and Ccr4-Not protein complexes) might be involved in the coupled evolution of RNA transcription and decay in yeast, a clear demonstration of the first scenario (Dori-Bachash et al, 2011). However, here in our system, to what extent the two scenarios account for the coordinated evolution of RNA transcription and decay awaits future functional studies.

*cis*-Divergence in RNA decay should result solely from the sequence variants on the mRNA transcripts affecting *cis*-regulatory elements (e.g., miRNA-binding sites). Therefore, it would be possible to investigate the regulatory mechanisms underlying the *cis*-divergence in RNA decay by analyzing the sequence differences of ASD genes between the two alleles. Indeed, by such analysis, we demonstrated that sequence variants affecting miRNA binding could contribute to the observed ASD divergence. In contrast, in our previous analysis of allele-specific translation efficiency using the same F1 cells, we did not observe the significant impact of miRNA binding on translation, indicating, at least in the cellular system as used in our studies, that miRNAs regulate gene expression mostly through RNA degradation (Hou et al, 2015). In addition to miRNA-binding sites, our sequence analysis also revealed that variants affecting RNA secondary structures could also lead to the *cis*-divergence in RNA decay. Interestingly, in contrast to miRNA-binding sites, we did not observe between the two alleles the significant correlation (or anti-correlation) between the stability of RNA secondary structure and the rate of RNA decay (Fig S16). This might

reflect the fact that different double-strand RBPs could either accelerate or decelerate RNA decay. For example, it has been shown that Staufen1 could bind to RNA duplexes and trigger the degradation of the bound RNAs (Park & Maquat, 2013), whereas HNRPA2B1 could bind to specific RNA secondary structures and thereafter stabilize the host transcripts (Hamilton et al, 1999).

Surprisingly, we did not find any significant impact of several known *cis*-regulatory features on the observed allelic biases in RNA decay, such as ARE and codon usage. A possible explanation is that ASD might be due to the combined effects of a large set of diverse mechanisms, and the individual contributions of these specific features with lower frequencies and/or smaller effect sizes might not be sufficient to reach statistical significance.

Finally, this study served as a first proof-of-principle investigation that used a mammalian F1 hybrid system to globally analyze the *cis*-divergences of RNA decay. One caveat of this study is that the conclusions were drawn from the results observed in mouse fibroblast cells. Thus, one future research direction would be to investigate whether our observations would remain the same in other mammalian tissues and cells. Furthermore, it has been shown that RNA decay plays more important roles during the response to extrinsic or intrinsic stimuli. Thus, future studies using our F1 system under those dynamic conditions would reveal more novel insights into the molecular mechanisms underlying the evolution of RNA decay in mammals.

# Materials and Methods

### F1(B×S) hybrid mouse fibroblast cell cultures

The F1(B×S) hybrid mice were obtained as described before (Gao et al, 2013). Adult mouse fibroblast cells were isolated and cultured according to the protocol from Encyclopedia of DNA Elements project (https://genome.ucsc.edu/encode/protocols/cell/mouse/Fibroblast_Stam_protocol.pdf) with modification of cell culture medium (RPMI 1640 Medium, GlutaMAX Supplement [Gibco; Life Technologies] with 0.5% FBS and 1% Penicillin/Streptomycin Solution).

### Actinomycin D treatment and RNA sequencing

Actinomycin D (10 mg/ml, Sigma-Aldrich) was directly added to cell cultures. Cells were collected at 0, 0.5, and 1.5 h after the addition of actinomycin D. Total RNA from the collected cell samples was extracted using TriZOL reagent (Life Technologies) following the manufacturer's protocol. Stranded mRNA sequencing libraries were prepared with 500 ng total RNA according to the manufacturer's protocol (Illumina). The libraries were sequenced in a 2 × 100 +7 manner on a HiSeq 2000 platform (Illumina).

### Reference sequences and gene annotation

The reference sequences and the Ensembl gene annotation of the C57BL/6J genome (mm10) were downloaded from the Ensembl FTP server (http://ftp.ensembl.org, version GRCm38, release 74). The RefSeq gene annotation was downloaded from the University of

California, Santa Cruz (UCSC), genome browser (http://hgdownload.soe.ucsc.edu/goldenPath/mm10/database/). The single nucleotide variants and indels between BL6 and SPRET were downloaded from the Mouse Genome Project Web site (http://www.sanger.ac.uk/). The vcf2diploid tool (version 0.2.6) in the AlleleSeq pipeline was used to construct SPRET genome by incorporating the single nucleotide variants and indels into BL6 genome (Rozowsky et al, 2011). The chain file between the two genomes was also reported as an output, which was further used with the UCSC liftOver tool. The liftOver tool from the UCSC Genome Browser (Kuhn et al, 2013) was applied to get SPRET gene annotation.

### Allele-specific sequencing read alignment

Flexbar was first used to trim RNA-seq reads that pass the Illumina filter to remove Illumina adapter sequences with parameters -x 6 -u 0 -m 50 -ae RIGHT -at 3 (Dodt et al, 2012). Read pairs that were concordantly mapped to the reference sequences of rRNA, tRNA, snRNA, snoRNA, and miscRNAs (available from Ensembl and RepeatMasker annotation) using Bowtie2 (version 2.1.0) with default parameters (in end-to-end and sensitive mode) were excluded.

The remaining reads were then aligned to the mouse genome reference sequences (see above) using TopHat (version 2.0.8) with default mapping parameters and Ensembl gene annotation (Trapnell et al, 2009). Concordantly mapped read pairs (i.e., mates of a read pair mapped to the same transcript with opposite orientation) were then assigned to the parental allele with less mapping edit distance; read pairs with equal edit distance to either allele were assigned as "common." Read pairs that mapped to sex chromosomes and mitochondrial DNA were excluded for further analysis. Genomic alignment coordinates for reads from the SPRET/EiJ allele were then converted to the corresponding locations in the C57BL/6J reference genome using the UCSC liftOver tool and their chain files.

### Filtering of SNP loci with potential allelic mapping and assignment biases

To estimate ASD, only the reads that could be unambiguously assigned to SNP loci from either allele were counted (see above). To avoid bias due to the potential misalignment of reads to the wrong allele, we used previously published datasets generated from fibroblast cell lines of the two parental strains (Gao et al, 2015). Specifically, we first created a mock F1 hybrid RNA-seq dataset by combining equal amounts of RNA-seq reads derived from the parental strains. We then performed the same alignment analysis as described above on the mock F1 hybrid and the two parental strain datasets. For each SNP locus, the numbers of reads assigned to the parent strains (in the original datasets) or specifically to the parental alleles (in the mock datasets) were then counted and compared and Fisher's exact test was used to filter the SNP loci with potential bias ($P$-value < 0.05, after Benjamini–Hochberg correction for multiple testing).

Because of potentially incomplete annotation of SNPs at paralogous genes or pseudogenes in the SPRET/EiJ genome, some reads, which could be mapped to multiple gene loci if the C57BL/6J sequence was used as a reference, were mapped to a unique position in the SPRET/EiJ genome. In such cases, removal of multiple mapped reads

(only from C57BL/6J allele) could lead to inaccurate calculation of ASD. To avoid such bias, for each SNP locus, based on the mock datasets, we compared the ratio of allele-specific reads, including multiple mapped reads, with that counting only uniquely mapped reads. Fisher's exact test was used to filter the SNP loci with potential bias ($P$-value < 0.05, after Benjamini–Hochberg correction for multiple testing).

### Estimation of allelic differences in mRNA decay rate

After SNP loci filtering (see above), only the genes with at least five SNPs supported by sufficient allelic reads in all different time course samples (i.e., $mRNA_{0\ h,\ BL6} + mRNA_{0\ h,\ SPRET} \geq 10$ and $mRNA_{0.5\ h,\ BL6} + mRNA_{0.5\ h,\ SPRET} \geq 10$ and $mRNA_{1.5\ h,\ BL6} + mRNA_{1.5\ h,\ SPRET} \geq 10$ and $mRNA_{0\ h,\ BL6} + mRNA_{0.5\ h,\ BL6} + mRNA_{1.5\ h,\ BL6} \geq 15$ and $mRNA_{0\ h,\ SPRET} + mRNA_{0.5\ h,\ SPRET} + mRNA_{1.5\ h,\ SPRET} \geq 15$) were considered for further analysis.

To determine whether a gene exhibited allelic differences in mRNA decay rate, we combined a previously published logistic model and a bootstrapping strategy (Andrie et al, 2014; Muzzey et al, 2014). Specifically, we let $N_i(t)$ be the number of mRNA transcripts for allele $i$ ($i$ = 1, 2, representing BL6 and SPRET) at time $t$. We assumed an exponential decay $\frac{dN_i(t)}{dt} = -\lambda dN_i(t)$ for a constant $\lambda$, such that $N_i(t) = N_i(0) \exp(-\lambda t)$. For each time point $t$, the number of RNA-seq reads that we can assign to an allele $n_i(t)$ is a fraction $f(t)$, of the total number of mRNA transcripts for that allele, such that $n_i(t) = f(t)N_i(t)$. We then assumed the model $n_i(t) \sim Poisson[f(t)N_i(t)]$. Under this model, the distribution of the counts for strain 1 (BL6) given the total is binomial:

$$p(t) = \frac{f(t)N_1(t)}{f(t)N_1(t) + f(t)N_2(t)} = \frac{\frac{N_1(0)}{N_2(0)}\exp(-[\lambda_1 - \lambda_2]t)}{\frac{N_1(0)}{N_2(0)}\exp(-[\lambda_1 - \lambda_2]t) + 1}.$$

Taking the log it gives:

$$\log\left(\frac{p(t)}{1-p(t)}\right) = \log\left(\frac{N_1(0)}{N_2(0)}\right) - [\lambda_1 - \lambda_2]t = \alpha + \beta t.$$

In this linear logistic model, the mRNA decay rate differences between the two alleles can be directly estimated using the parameter $\beta$. The parameter $\exp(\beta)$ represents the change in the odds of observing an mRNA allele of the strain 1 type, given a 1-h increase in time ($t$ is measured in hours). If decay rates are the same in both strains ($\lambda_1 = \lambda_2$), then $\beta = 0$.

To assess the uncertainty of estimated mRNA decay rate differences, a bootstrapping procedure was applied (Muzzey et al, 2014). Specifically, for each gene consisting of a list of $n$ ($n \geq 5$) SNP loci, we generated 5,000 new lists, each consisting of $n$ SNP loci that were chosen at random with replacement from the original list. For each of the 5,000 random lists, mRNA decay rate differences between the two alleles were estimated using the above logistic model, and then yielded a bootstrap distribution, from which we got the bootstrapping mean and standard deviation. To determine the statistical significance of genes with ASD, we calculated a $P$-value based on the $z$-score that represented how many folds of standard deviation the bootstrapping mean deviated from zero. The raw

$P$-values were then adjusted using the Benjamini–Hochberg method. To estimate the FDR, we used a similar permutation strategy as described before (Sterne-Weiler et al, 2013). In brief, gene labels were shuffled for 100 times in both replicates, and in each of the 100 shuffled sets, we calculated the number of genes in both replicates meeting the bootstrapping significance requirement (adjusted $P$-value < 0.05) and decay rate difference requirement ($\beta = |\lambda_1 - \lambda_2| > x$), and biased toward the same allele. Then, for each of the 100 permutations of each value $x$, the FDR was estimated as false positives divided by the number of real genes passing the same threshold. Finally, Benjamini–Hochberg–adjusted $P$-value < 0.05 and $|\Delta\lambda| > 0.06$ in both replicates (FDR = 4.18%) was used as the threshold for determining whether a gene exhibited significant allelic differences in mRNA decay rate.

### PacBio sequencing and data analysis

Starting with 500 ng of total RNA, DNase treatments were first performed according to the manufacturer's protocol (TURBO DNA-free kit; Thermo Fisher Scientific) for samples collected at 0 and 1.5 h after actinomycin D treatment. Reverse transcription (RT) reactions were followed using random hexamer primers (Thermo Fisher Scientific) and SuperScript II reverse transcriptase (Thermo Fisher Scientific). PCR reactions were then performed using 1 μl of RT products as template in 50 μl of GoTaq PCR system (Promega). PCR primers were designed for amplifying the genic region containing sequence variants between B6 and SP transcripts. All primer sequences are listed in Table S2. The PCR program was as follows: 4 min at 95 °C; followed by 28 cycles of 30 s at 95 °C, 30 s at 55 °C, and 45 at 72 °C; and a final elongation of 10 min at 72 °C. Different PCR products from the same RT product using different primers were then mixed and purified using Agencourt AMPure XP system (Beckman Coulter) and quantified by Qubit HS dsDNA measurement system (Life Technologies). These mixed PCR products were then sequenced on PacBio RS SMRT platform according to the manufacturer's instruction.

Sequence reads from the PacBio RS SMRT chip were processed through PacBio's SMRT-Portal analysis suite to generate circular consensus sequences. The circular consensus sequences were then mapped to a reference database containing both alleles of target genes using BLAST with default parameters. The best hit was retained for each aligned sequence read. These reads were then assigned to C57BL/6J or SPRET/EiJ allele with fewer mismatches. The numbers of reads assigned to either allele of each gene at 0 and 1.5 h were counted, respectively. The following equation was used to estimate ASD:

$$ASD = \log 2\left(\frac{mRNA_{1.5\ h,\ BL6}/mRNA_{1.5\ h,SPRET}}{mRNA_{0\ h,BL6}/mRNA_{0\ h,SPRET}}\right).$$

### Selection of control genes without ASD

To compare with the genes exhibiting ASD, we selected a separate group of control genes that were also supported by sufficient allelic reads (see above) but did not show any difference in decay rate between the two alleles: 1) $P$-value from bootstrapping analysis >0.05 for both replicates; 2) $|\Delta\lambda| < 0.03$ for both replicates; 3)

bootstrapping deviation <0.1 (i.e., 95% quantile of all genes) for both replicates.

To analyze the sequence features of the genes exhibiting ASD, we further selected a subset of these control genes, which possessed similar density of sequence variants as ASD genes, to avoid the potential bias due to the different variant densities between ASD and control genes. Specifically, based on the distribution of sequence variant density across the whole transcript in ASD genes, we randomly selected from all the control genes a subset with the same variant density distribution as ASD genes.

### Local RNA secondary structure

Local RNA secondary structure MFE was calculated using *RNAfold* from *ViennaRNA* package version 2.1.9 with default parameters at a temperature of 37 °C (Lorenz et al, 2011). Specifically, we calculated the MFE of an *i*-nt region (*i* = 21, 41, 61, 81, and 101) flanking each SNP between C57BL/6J allele and SPRET/EiJ allele. The variant of interest was placed at the center of each window if the whole window was within the transcript; however, if the variant was <($i$ − 1)/2 nt (e.g., 20 bp for *i* = 41) from the end of the RNA transcript, the *i*-nt window was shifted such that its boundary lay at the end of the transcript. We then calculated the absolute difference of MFE between the two alleles (|ΔMFE|) for each SNP. For each gene or each region (including 5′ UTR, CDS, and 3′ UTR), we used the maximum |ΔMFE| among all its SNP loci to represent the allelic difference in mRNA secondary structure.

### Other sequence features, including miRNA-binding sites, codon usage bias, and AREs

miRNA target sites in each gene were counted using a custom Perl script by matching three site-types (i.e., 8mer, 7mer-m8, and 7mer-1A) using *TargetScan* v7 (Friedman et al, 2009) as previously described (Hou et al, 2015). For both control and ASD gene groups, we used only the genes with ≥10 miRNA-binding sites combining the two alleles together and ≥1 different sites between the two alleles.

The effects of AREs were estimated using the *AREScore* algorithm with default parameters (Spasic et al, 2012). Briefly, *AREScore* calculates a score based on the number of AUUUA pentamers, the distance between these pentamers, and whether they are located within an AU-block. The 3′ UTR sequences for either allele of each gene were submitted to *AREScore* web server (http://arescore.dkfz.de/arescore.pl).

Codon usage bias was estimated using the CAI calculated using *CodonW* version 1.4.4 (http://codonw.sourceforge.net/). For each gene, the coding sequence (CDS) of each allele was used as input for *CodonW*.

In the analysis of these sequence features, the difference between the stable (slow-decaying) and unstable (fast-decaying) alleles was calculated for ASD and control genes (see the selection of control genes without ASD section for details) separately. RefSeq annotation was used to separate each coding gene into 5′ UTR, CDS, and 3′ UTR. For the genes with multiple isoforms, the longest one was used. When 5′ UTR (CDS and 3′ UTR) region is considered, only the genes with 5′ UTR (CDS and 3′ UTR) are used. Note that some genes do not have annotated 5′ UTR or 3′ UTR, and noncoding RNAs are not considered when separating genes into different regions.

### Data Availability

The RNA-seq data from this publication have been submitted to the European Nucleotide Archive (http://www.ebi.ac.uk/ena) and assigned the accession no. ERP017147.

## Supplemental Information

## Acknowledgements

We thank Dr. Xi Wang for sharing the script for miRNA analysis and helpful discussions. W Chen was supported by National Natural Science Foundation of China (31771443), basic research grant from Science and Technology Innovation Commission of Shenzhen Municipal Government (JCYJ20170307105752508), China Thousand Talent Program, startup funds from Southern University of Science and Technology, and Peacock Plan of Shenzhen Municipal Government. W Sun and Q Gao were supported by the Chinese Scholarship Council.

### Author Contributions

W Sun: conceptualization, validation, methodology, writing—original draft, review, and editing, experiments.
Q Gao: conceptualization, resources, software, methodology, and writing—original draft, review, and editing.
B Schaefke: data curation and writing—original draft, review, and editing.
Y Hu: data curation, formal analysis, and writing—original draft, review, and editing.
W Chen: conceptualization, supervision, writing—original draft, project administration, and writing—review and editing.

### Conflict of Interest Statement

The authors declare that they have no conflict of interest.

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
