## [Reviewer comments · Life Science Alliance]

Pervasive allele-specific regulation on RNA decay in hybrid mice

Wei Sun^{2,1,4}, Qingsong Gao^{2,4}, Bernhard Schaefer¹, Yuhui Hu¹, Wei Chen
DOI: 10.26508/lisa.201800052

Review timeline:	Submission Date:	13 March 2018
	1 st Editorial Decision:	4 April 2018
	1 st Revision Received:	22 April 2018
	2 nd Editorial Decision:	30 April 2018
	2 nd Revision Received:	2 May 2018
	Accepted:	3 May 2018

Report:

(Note: Letters and reports are not edited. The original formatting of letters and referee reports may not be reflected in this compilation.)

1st Editorial Decision 4 April 2018

Thank you for submitting your manuscript entitled "Pervasive allele-specific regulation on RNA decay in hybrid mice" to Life Science Alliance. The manuscript was assessed by expert reviewers, whose comments are appended to this letter. We invite you to submit a revision if you can address the reviewers' key concerns.

As you will see, the reviewers state that the work is nicely complementary and confirms with a robust and interesting dataset previous notions. The issues raised by the reviewers seem straightforward to address, and I would thus like to ask you to provide a revised version of your work. Please note that the further reaching insight that could be provided as stated by reviewer #1 is not mandatory for publication in Life Science Alliance.

Thank you for this interesting contribution to Life Science Alliance. We are looking forward to receiving your revised manuscript.

REFeree REPORTS

Reviewer #1 (Comments to the Authors (Required)):

In this manuscript Sun et al. measure allele specific mRNA decay rates in a mouse hybrid F1 cell line. For that the authors inhibit transcription with ActD and measure mRNA abundance by RNA-Seq at different time points. By studying mRNA regions containing SNPs the authors compare differences in allele-specific RNA decay with differences in mRNA abundance. The authors also study the association of allele-specific mRNA decay differences with known players in RNA decay such as miRNA or secondary structure.

This paper complement previous studies in yeast and eQTL studio in human cell lines, and technically seems well performed. This is a descriptive paper where the authors produce an interesting dataset to confirm previous knowledge. The authors perform a technical validation of their accuracy of the measurement with PacBio, but not a functional validation of the effect of the SNPs in stability. Specially interesting would be the functional implication of those differences, for example on the ability of the different alleles to adjust their total mRNA abundance to new

conditions in an induction time course. Or the exploration of the behaviour of those allele with different stability in previously published datasets. Of course, those analysts could be included in a future work and is mainly an editorial decision. I have the following questions and recommendations:

- 1) The authors should expand the discussion on their ability to measure accurately allele specific decay rates vs abundance. Even if the FDR applied is the same, features more difficult to measure (noisier) will be more difficult to call as significantly different. Could the increased noise associated to the measure of decay rates explain part of the discrepancy with abundance? Specifically, up to what degree the differences that the authors claim (change in stability but not in abundance) could be explain by a technical limitation on the measure?
- 2) In Fig 3A, can the increase presence of SNPS in the differential alleles be due to a selection bias? Meaning, alleles with more SNPs more likely to be called significantly different?
- 3) How does the different stability measured here relate to the differential translation measured by the authors a few years back in the same system? (Hou et al 2015 MSB)

Reviewer #2 (Comments to the Authors (Required)):

Sun et al. have investigated, using a B6/SPRET F1 hybrid mouse system, allele-specific regulation of RNA decay. Furthermore, they demonstrate that allele-specific decay effects are often countered by allele-specific transcription effects so as to provide equivalent mRNA allelic levels. Their study is complementary to the QTL approach of Pai et al., 2012, and thus corroborates their findings.

The study's methods have been performed to a high level, its results have been interpreted appropriately, and the manuscript is very well written. I have few comments and questions:

1. p8. The assumption of exponential decay is likely to be appropriate, but should be explored and the decay curves plotted as in Pai et al., 2012 (Figure 1a).
2. p10. miRNA recognition elements. Are these predicted using TargetScan v7? I would be interested whether the same results are found using miRanda.
3. p11 Similarly, I would be interested in whether the MFE results are robust to a different RNA secondary structure prediction algorithm.
4. The Discussion should say that these are results that are specific to mouse fibroblasts and thus may not be more generally reflective of other mammalian tissues or cells. Explanation 3 on p16 would be consistent with the vastly greater effective population size of yeasts relative to mammals.
5. What genes (GO Terms) are enriched in rapidly versus slowly decaying transcripts? Long or short 3'UTRs. Compare with Pai et al., 2012.

Minor comments.

1. p9 and elsewhere. The exponent in $2.0e-11$ should, instead, be 2.0×10^{-11} .
2. In Figures, explain the meaning of the asterisks (*, **, ***).
3. Cell cultures. Are the duplicates from 2 mice, or else duplicates of cells from 1 mouse?
4. p22 "sex chromosomes".
5. p24. Equations. Given the explanation given for $\exp(\beta t)$ then explain that t is in minutes.

Reviewer #3 (Comments to the Authors (Required)):

In this manuscript Sun et al. explore the phenomenon of cis regulatory effects on RNA decay using hybrid mice. This has been done previously in other model organisms, such as yeast, however the authors state that such an analysis has not been performed using mammalian model systems. The authors analyze allelic count data at heterozygous variants that tag each of the parental strains over a time course after blocking transcription, and observe numerous genes with significant allelic effects on RNA stability. By analyzing genetic variation on each alleles independently in genes that show significant effects they identify an abundance of genetic variation as compared to control genes, and further analyses implicate regulation by miRNA and RNA secondary structure as potential mechanisms. Finally, the authors find that that majority of genes displaying significant allele

specific abundance do not have significant allele specific decay, suggested that most cis regulatory effects on abundance result from modulating transcription. On the other hand, the authors observe that most genes with significant allele specific RNA decay do not exhibit significant allele specific abundances, suggested the widespread prevalence of potential compensatory regulatory effects.

Overall, I found the topic of the research to be of general interest, however as I am not an expert in the field of RNA stability, I can't comment on how novel the findings are with respect to what is already known. The general idea for the analysis (analyzing allelic effects in F1 hybrids) is sound, and it seems that the authors have given sufficient thought to potential issues that arise when using allelic count data (e.g. potentially mapping bias). I do however have concerns related to the analyses that were performed, which are important to address in order to justify the conclusions made by the authors. I expect that these concerns could be addressed in a revised manuscript. Below please find my comments.

General Manuscript Comments

- Are the unstable / stable alleles distributed evenly between the two parental strains, or is there is a bias towards one or the other being stable/unstable?
- It would be clearer if the authors listed p-values on the figures instead of in the figure legends.
- In the discussion change "inter-human-individual" to "human inter-individual".

Specific Analysis Comments

Selection of control genes for ASD analyses (Figure 3):

- Of the 8,815 genes with sufficient allelic reads, only 1,319 confidently show no evidence of ASD. From this number it seems that the authors are potentially being very stringent in terms of their requirement for control genes.
- For analysis of sequence features in ASD genes, the authors indicate that they selected control genes "which possessed similar density of sequence variants as ASD genes". The authors need to quantitatively define what they qualified as a similar density. They should also indicate the final number of control genes that match this criterion.

Figure 3a:

- One potential confounder for the analysis presented here is that genes with more variants, and thus potentially more allelic read coverage are better powered to detect ASD. This issue would persist regardless of the minimum sufficient allelic read count requirement. The authors need to control for differences in allelic read coverage between control and ASD genes for this analysis in particular.

Figure 3b:

- From the figure panel it appears that not all of the ASD / control genes were used. I believe this is because only genes with at least one predicted miRNA binding site were used, but this should be mentioned in the figure legend.
- In the figure legend the plots are referred to as "Barplots", this should be "Boxplots".
- For the control genes used, were only a subset of the genes that had similar variant density used in order to match the number of ASD genes? If so, the authors need to explain how this subsetting was done and demonstrate that their result is robust to which control genes were selected.
- The randomization of the haplotypes in the control genes to be a bit problematic, although I do understand that you can't assign one haplotype as being unstable or stable. An additional control could be to look at the difference in number of miRNA binding sites by defining haplotypes by their parental origin (BL6/SPRET). It could be for example, that one mouse strain overall has more miRNA binding sites than the other, and that this is not limited to genes where ASD is observed.

Figure 3c:

- The same concerns about selection of control genes mentioned for Figure 3b apply here.

Figure 4:

- When defining the minimum fold change requirement for defining ASA significance, the manuscript text states that the same threshold as for ASD was used: "Using the same bootstrapping strategy on \log_2 fold change of allelic expression at the same threshold". It's not entirely clear to me how the $|\log_2(\text{fold change})| > 1$ threshold correlates to the $\Delta \lambda > 0.06$ threshold in terms of effect size. This is important because the level at which this cutoff is defined will directly affect the authors' conclusion that "among the 621 ASD genes, most (516 genes, 83.1%) didn't exhibit significant ASA", which is a central finding of their work. As such, I think the authors need to come up with a more robust analysis framework that is not dependent on setting arbitrary cutoffs for what constitutes significant ASA and then simply looking at gene list overlaps.
- It would be more informative to analyze the difference in effect size magnitude between ASA and ASD to produce a more quantitative estimate of how much ASA is observed relative to ASD. For example, the authors could quantify the amount of ASA ($|\log_2(\text{fold change})|$) observed in genes with significant ASD and compare that to control genes without ASD to get an idea of relative increase or decrease in AST affects that could potentially be compensatory to ASD effects at those genes. It would also be interesting to see if overall there is a correlation between ASD and ASA effect size.
- From the plot it appears that everything with $|\log_2(\text{RNABL6/RNASPRET})| > 1$ has significant ASA? Is this really the case? Surely after multiple testing correction, there would be some genes where the effect size was sufficiently large, but due to other reasons, e.g. low overall read count, that the difference is not significant.

Reviewer #1:

In this manuscript Sun et al. measure allele specific mRNA decay rates in a mouse hybrid F1 cell line. For that the authors inhibit transcription with ActD and measure mRNA abundance by RNA-Seq at different time points. By studying mRNA regions containing SNPs the authors compare differences in allele-specific RNA decay with differences in mRNA abundance. The authors also study the association of allele-specific mRNA decay differences with known players in RNA decay such as miRNA or secondary structure

This paper complement previous studies in yeast and eQTL studio in human cell lines, and technically seems well performed. This is a descriptive paper where the authors produce an interesting dataset to confirm previous knowledge. The authors perform a technical validation of their accuracy of the measurement with PacBio, but not a functional validation of the effect of the SNPs in stability. Specially interesting would be the functional implication of those differences, for example on the ability of the different alleles to adjust their total mRNA abundance to new conditions in an induction time course. Or the exploration of the behaviour of those allele with different stability in previously published datasets. Of course, those analysts could be included in a future work and is mainly an editorial decision. I have the following questions and recommendations:

- 1) The authors should expand the discussion on their ability to measure accurately allele specific decay rates vs abundance. Even if the FDR applied is the same, features more difficult to measure (noisier) will be more difficult to call as significantly different. Could the increased noise associated to the measure of decay rates explain part of the discrepancy with abundance? Specifically, up to what degree the differences that the authors claim (change in stability but not in abundance) could be explain by a technical limitation on the measure?

Answer: To avoid the effect of arbitrary thresholds, we used different combinations of FDRs for ASA and ASD (0.005, 0.0075, 0.01, 0.025, 0.05, 0.075 and 0.1 for ASD and ASA, respectively). As shown in Figure R1 below, the conclusion is consistent across different thresholds. For example, when we relaxed the FDR threshold to 0.1, 706 and 3,016 genes showed significant ASD and ASA, respectively. 2,747 out of 3,016 genes with significant ASA (91.1%) didn't exhibit significant ASD. On the other hand, 437 out of 706 ASD genes (61.9%) didn't exhibit significant ASA.

This figure is added to the revised manuscript as supplementary Figure S13.

Figure R1. The role of ASD in the allelic difference of RNA abundances under different combinations of FDR thresholds. Barplots showing the percentage of ASD genes in those with significant ASA (blue bars) and the percentage of ASA genes in those with significant ASD (red bars) at different combinations of FDR thresholds (0.005, 0.0075, 0.01, 0.025, 0.05, 0.075 and 0.1 for ASD and ASA, respectively).

2) In Fig 3A, can the increase presence of SNPS in the differential alleles be due to a selection bias? Meaning, alleles with more SNPs more likely to be called significantly different?

Answer: We thank the referees for pointing this out. We believed that the higher variant density in the ASD genes reflected the *cis*-variants causing stability divergence, although this could also be due to the higher detection power from genes with high variant density, as the two referees suggested. However, as referee 3 rightfully pointed out, for the latter, what determined the detection power is allelic read counts, which in principle could correlate with the variant density. To disentangle the two scenarios, we compared the allelic read count distributions between ASD genes and control genes. As shown in the Figure R2 below, the allelic read counts are higher in control genes than ASD genes, for both the measurement at 0 hr of Actinomycin D treatment (Figure R2 A) and at all three time points of Actinomycin D treatment (Figure R2 B). Therefore, we think the observed high variant density in the ASD genes indeed reflects the *cis*-variants.

Figure R2. Allelic read counts distribution in ASD and controls genes. Violin plot comparing the allelic read counts distribution (log₂ transformed, y-axis) between ASD genes (red) and controls (blue) using the reads at 0 hr of Actinomycin D treatment (A, $t=0$ h) or all three time points (B, $t=0$, 0.5 and 1.5h). Allelic read count in controls genes are significantly higher than that in ASD genes (P -value= 1.6×10^{-13} and 2.2×10^{-16} , respectively)..

3) How does the different stability measured here relate to the differential translation measured by the authors a few years back in the same system? (Hou et al 2015 MSB)

Answer: We compared the allele-specific translational efficiency (ASTE) with allele-specific degradation for 6,391 genes with both datasets available. In the figure below, x-axis and y-axis represent allelic decay rate difference ($\Delta\lambda$) and log₂-transformed fold change of translational efficiency between two alleles. Grey dash lines indicate twofold change for translation efficiency (same cutoff as Hou et al, 2015, MSB) and 0.06 for decay rate difference, respectively (FDR<0.05). Genes with significant allelic bias at only translational level, only decay level and both levels were depicted in orange, green and blue, respectively. As shown in this figure, most of the 852 genes with significant ASTE (784 genes, 92.0%) didn't exhibit significant ASD. On the other hand, most of the significant ASD genes (435 out of 503 genes, 86.5%) didn't exhibit significant ASTE. These results suggest that in contrast to the potential coupling of decay and transcription, there is no mechanistic link between the evolution of RNA decay and mRNA translation.

Figure R3. Relationship between allele-specific degradation and translation efficiency. Scatterplot comparing each gene's allele-specific translation efficiency (log₂-transformed fold change at y-axis) and decay ($\Delta\lambda$ at x-axis). Grey dash lines indicate twofold change for translation efficiency and 0.06 for decay rate difference, respectively (FDR<0.05). Genes with significant allelic bias at only translation efficiency level, only decay level and both levels were depicted in orange, green and blue, respectively.

Reviewer #2:

Sun et al. have investigated, using a B6/SPRET F1 hybrid mouse system, allele-specific regulation of RNA decay. Furthermore, they demonstrate that allele-specific decay effects are often countered by allele-specific transcription effects so as to provide equivalent mRNA allelic levels. Their study is complementary to the QTL approach of Pai et al., 2012, and thus corroborates their findings.

The study's methods have been performed to a high level, its results have been interpreted appropriately, and the manuscript is very well written. I have few comments and questions:

1. p8. The assumption of exponential decay is likely to be appropriate, but should be explored and the decay curves plotted as in Pai et al., 2012 (Figure 1a).

Answer: Please see Figure R4 below. For this analysis, we ignored the sequence variants between the two strains and mapped the F1 hybrid reads to the standard mouse genome (B6) using Tophat2. Then Cufflinks was used to estimate the FPKM value for each gene in each sample from different time points. For each gene, the mean value of the two replicates was used. As shown in the figure, for the majority of genes, the decay fits with the exponential model.

Figure R4. Profiles of decay rates. Distribution of genome-wide decay profiles across the timecourse experiment (x-axis), where each decay curve shows the decrease in gene expression level (y-axis) relative to the steady state. Each line represents the gene-specific decay profile.

2. p10. miRNA recognition elements. Are these predicted using TargetScan v7? I would be interested whether the same results are found using miRanda.

Answer: Yes, miRNA binding sites in the manuscript were predicted using TargetScan v7. As suggested, miRanda was used to validate the findings.

Specifically, miRanda (v3.3a) was downloaded from <http://34.236.212.39/microrna/getDownloads.do>. Mature miRNA sequences (fasta format) were downloaded from miRBase (<http://www.mirbase.org/>). miRanda was run with the following parameters <miranda mirna.fa target.fa -sc 180 -en 1 -scale 4>. Same as TargetScan, only the genes with at least one SNPs on the predicted miRNA binding sites and ≥ 10 miRNA binding sites combining the two alleles together were considered. As shown in Figure R5 below, we could still observe that the unstable alleles possess more miRNA target sites than the stable alleles for ASD genes, but no significant difference exists for the control group.

Figure R5. Boxplots and scatterplots showing the distribution of miRNA binding site number difference between the stable and unstable alleles for genes with significant ASD and controls estimated using miRanda. For controls, the difference centered around zero (P -value=0.77, two-sided Mann-Whitney U test), whereas in ASD genes, unstable alleles tend to possess more miRNA target sites than the stable alleles (P -value=0.027, two-sided Mann-Whitney U test). Only the genes with ≥ 10 miRNA binding sites combining the two alleles together and ≥ 1 different sites were used.

3. p11 Similarly, I would be interested in whether the MFE results are robust to a different RNA secondary structure prediction algorithm.

Answer: To check whether the results are robust to a different RNA secondary structure prediction algorithm, RNAstructure (v6.0.1, <http://rna.urmc.rochester.edu/RNAstructure.html>, PMID: 23620284) was used with the following parameters <Fold 41-bp-window.fa output -MFE>.

Specifically, for each sequence variant, we calculated the minimal free energy (MFE) of a 41bp-RNA segments (20 nt flanking each variant) along the whole transcript for the two alleles separately, and then calculated their absolute difference. For each gene, we used the maximum $|\Delta\text{MFE}|$ among all the variants to represent the allelic difference in mRNA secondary structure. As shown in Figure R6 below, we still observed significantly larger allelic differences of $|\Delta\text{MFE}|$ values in ASD genes compared to the control genes.

Figure R6. Violin plots and scatterplots comparing the distribution of the absolute minimal free energy difference (Δ MFE) between ASD genes and controls. The horizontal lines indicate the median. Compared to controls, ASD genes exhibited larger allelic differences (P -value=0.042, two-sided Mann-Whitney U test).

4. The Discussion should say that these are results that are specific to mouse fibroblasts and thus may not be more generally reflective of other mammalian tissues or cells. Explanation 3 on p16 would be consistent with the vastly greater effective population size of yeasts relative to mammals.

Answer: We thank the referee for the suggestions, and we modified the manuscript accordingly as following:

p20, last paragraph of the Discussion section - “Finally, this study served as a first proof-of-principle investigation that used a mammalian F1 hybrid system to globally analyze the cis-divergences of RNA decay. One caveat of this study is that the conclusions were drawn from the results observed in mouse fibroblast cells. Thus one future research direction would be to investigate whether our observation would remain the same in other mammalian tissues and cells. Furthermore, it has been shown that RNA decay plays more important roles during the response to extrinsic or intrinsic stimuli. ...”

p16, explanation 3 – “3) Gene expression regulation in yeasts and mammals evolved along different trajectories, with stronger stabilizing selection in *Saccharomyces* than mammals. Such scenario would be consistent with the vastly greater effective population size of yeasts relative to that of mammals. 4) ...”

5. What genes (GO Terms) are enriched in rapidly versus slowly decaying transcripts? Long or short 3'UTRs. Compare with Pai et al., 2012.

Answer: To identify rapidly and slowly decaying transcripts, we ignored the sequence variants between the two strains and mapped the F1 hybrid reads to the standard mouse genome (B6) using Tophat2. Then Cufflinks was used to estimate the FPKM value for each gene in each sample. For each gene, the mean value of the two replicates was used. Instead of estimating decay rate, we simply calculated the ratio of gene expression at 1.5h divided by that at steady state, and then ranked all the genes based on this ratio in increasing order. In this case, all the rapidly decaying transcripts are at the top of the list while the slowly decaying ones are at the bottom. Then we selected top/bottom 100, 200, 300, 400, 500, 600, 700, 800, 900 and 1000 genes, and compared their 3'UTR length with all the expressed genes. For GO analysis, only the top/bottom 500 genes were used. GeneTrail2 (<https://genetrail2.bioinf.uni-sb.de/>) was used to identify the enriched terms. Similar to Pai et al., 2012, the tests were performed using all GO categories and KEGG pathways.

As shown in Figure R7 below, in general, the 3'UTR length of rapidly decaying (unstable) genes is significantly longer than average of all genes if top 500~1000 genes are used. In contrast, the 3'UTR length of slowly decaying (stable) genes is significantly shorter than average of all genes if top 200~1000 genes are used. This trend is consistent with Pai et al. 2012.

As to GO analysis, the result is also consistent with Pai et al. 2012 in general: rapidly decaying genes are enriched in those regulating gene expression, e.g. transcription factor complex, nucleolus, RNA polymerase II transcription factor activity sequence specific DNA binding, etc; slowly decaying genes are enriched in cellular and organelle-related processes, e.g. extracellular matrix, cell substrate adhesion, etc. The top five terms for GO categories and KEGG pathways are listed in the table below.

These results are important quality controls for this study. However, they are not the focus of this study, therefore not included in the revised manuscript.

Figure R7. Boxplots comparing 3'UTR length between rapidly/slowly decaying genes and all genes. The 3'UTR length of rapidly decaying (unstable) genes is significantly longer than average of all genes if top 500~1000 genes are used. In contrast, the 3'UTR length of slowly decaying (stable) genes is significantly shorter than average of all genes if top 200~1000 genes are used (NS.: not significant; *: P -value<0.05; ***: P -value<0.001).

**Rapidly decaying genes:**

GO - Cellular Component			
Name	Number of hits	Expected score	Adjusted p-value
transcription factor complex(4)	26	2.42825	1.43E-014
nucleolus(5)	28	4.85651	6.33E-010
nucleoplasm part(5)	26	4.09355	6.33E-010
chromatin(3)	23	3.32326	1.80E-009
transferase complex(5)	26	4.72446	7.05E-009
KEGG - Pathways			
Name	Number of hits	Expected score	Adjusted p-value
MAPK signaling pathway	20	1.77534	1.85E-011
TNF signaling pathway	14	0.784964	3.14E-010
FoxO signaling pathway	14	0.961031	2.57E-009
Hippo signaling pathway	14	1.07841	8.01E-009
HTLV-1 infection	17	1.94407	1.58E-008
GO - Biological Process			
Name	Number of hits	Expected score	Adjusted p-value
vasculature development(5)	48	4.35765	1.92E-028
blood vessel development(4)	47	4.18158	1.96E-028
negative regulation of cell differentiation(4)	48	4.59241	5.85E-028
positive regulation of cell death(4)	43	4.06421	5.01E-025
blood vessel morphogenesis(4)	40	3.514	2.33E-024
GO - Molecular Function			
Name	Number of hits	Expected score	Adjusted p-value
RNA polymerase II transcription factor activity sequence specific DNA binding	55	4.20359	1.44E-037
transcription regulatory region sequence specific DNA binding(8)	54	4.409	9.41E-036
sequence specific double stranded DNA binding(7)	54	4.55573	3.00E-035
RNA polymerase II regulatory region sequence specific DNA binding(9)	49	3.89548	5.69E-033
RNA polymerase II regulatory region DNA binding(8)	49	3.93949	7.41E-033

Slowly decaying genes:

GO - Cellular Component			
Name	Number of hits	Expected score	Adjusted p-value
extracellular matrix(2)	59	2.79506	9.15E-052
proteinaceous extracellular matrix(3)	50	2.3769	1.27E-043
extracellular matrix component(2)	27	0.931686	5.31E-026
adherens junction(4)	37	2.94178	9.25E-025
cell substrate adherens junction(4)	35	2.50895	9.25E-025
KEGG - Pathways			
Name	Number of hits	Expected score	Adjusted p-value
ECM-receptor interaction	20	0.616233	1.50E-019
Focal adhesion	23	1.45989	1.02E-016
PI3K-Akt signaling pathway	23	2.51629	4.10E-012
Alzheimer's disease	15	1.21779	2.72E-009
Protein digestion and absorption	12	0.623569	2.72E-009
GO - Biological Process			
Name	Number of hits	Expected score	Adjusted p-value
cell substrate adhesion(4)	32	2.00276	2.53E-022
extracellular matrix organization(5)	27	1.41587	1.46E-020
extracellular structure organization(4)	27	1.42321	1.46E-020
regulation of cell motility(4)	40	4.85651	2.78E-019
regulation of cell migration(5)	38	4.62909	3.22E-018
GO - Molecular Function			
Name	Number of hits	Expected score	Adjusted p-value
calcium ion binding(6)	55	3.73408	4.41E-040
sulfur compound binding(3)	28	1.59927	3.80E-021
growth factor binding(4)	18	0.872997	4.41E-014
glycosaminoglycan binding(4)	20	1.24714	4.52E-014
actin binding(5)	24	2.58965	1.70E-012

Minor comments.

1. p9 and elsewhere. The exponent in 2.0e-11 should, instead, be 2.0x10⁻¹¹.

Answer: All changes were made accordingly.

2. In Figures, explain the meaning of the asterisks (*, **, ***).

Answer: We directly listed the p-values in the new figures.

3. Cell cultures. Are the duplicates from 2 mice, or else duplicates of cells from 1 mouse?

Answer: The duplicates are duplicates of cells from the same mouse.

4. p22 "sex chromosomes".

Answer: "sexual chromosomes" was changed to "sex chromosomes".

5. p24. Equations. Given the explanation given for $\exp(\beta)$ then explain that t is in minutes.

Answer: The explanation of t was added – "The parameter $\exp(\beta)$ represents the change in the odds of observing an mRNA allele of the strain 1 type given a 1-h increase in time (t is measured in hours). "

Reviewer #3:

In this manuscript Sun et al. explore the phenomenon of cis regulatory effects on RNA decay using hybrid mice. This has been done previously in other model organisms, such as yeast, however the authors state that such an analysis has not been performed using mammalian model systems. The authors analyze allelic count data at heterozygous variants that tag each of the parental strains over a time course after blocking transcription, and observe numerous genes with significant allelic effects on RNA stability. By analyzing genetic variation on each alleles independently in genes that show significant effects they identify an abundance of genetic variation as compared to control genes, and further analyses implicate regulation by miRNA and RNA secondary structure as potential mechanisms. Finally, the authors find that that majority of genes displaying significant allele specific abundance do not have significant allele specific decay, suggested that most cis regulatory effects on abundance result from modulating transcription. On the other hand, the authors observe that most genes with significant allele specific RNA decay do not exhibit significant allele specific abundances, suggested the widespread prevalence of potential compensatory regulatory effects.

Overall, I found the topic of the research to be of general interest, however as I am not an expert in the field of RNA stability, I can't comment on how novel the findings are with respect to what is already known. The general idea for the analysis (analyzing allelic effects in F1 hybrids) is sound, and it seems that the authors have given sufficient thought to potential issues that arise when using allelic count data (e.g. potentially mapping bias). I do however have concerns related to the analyses that were performed, which are important to address in order to justify the conclusions made by the authors. I expect that these concerns could be addressed in a revised manuscript. Below please find my comments.

General Manuscript Comments

- Are the unstable / stable alleles distributed evenly between the two parental strains, or is there is a bias towards one or the other being stable/unstable?

Answer: The distribution of unstable/stable alleles is not significantly biased towards either strain. We address this question using two strategies:

-Figure R8 A below shows the percentage of genes with more stable B6 allele or more stable SPRET allele for the genes with ASD and all genes, respectively. The percentages are not significantly different from 0.5 for either group (P -value: 0.065 and 0.12 for ASD and all genes, respectively, Proportion test).

-Figure R8 B below shows the density plot of $\Delta\lambda$ for ASD (blue) and all genes. $\Delta\lambda$ is not significantly different from 0 (P -value=0.2 and 0.058 for ASD and all genes, respectively, t -test).

Figure R8. Distribution of stable/unstable alleles. (A) Barplots showing the percentage of genes with more stable B6 allele or more stable SPRET allele. The percentages are not significantly different from 50% (P -value=0.065 and 0.12 for ASD and all genes, respectively, Proportion test). (B) Density plot of $\Delta\lambda$ for ASD (blue) and all genes (red). $\Delta\lambda$ is not significantly different from 0 (P -value=0.2 and 0.058 for ASD and all genes, respectively, t-test).

- It would be clearer if the authors listed p-values on the figures instead of in the figure legends.

Answer: All the p-values are listed on the new figures.

- In the discussion change "inter-human-individual" to "human inter-individual".

Answer: The change has been made in the new version.

Specific Analysis Comments

Selection of control genes for ASD analyses (Figure 3):

- Of the 8,815 genes with sufficient allelic reads, only 1,319 confidently show no evidence of ASD. From this number it seems that the authors are potentially being very stringent in terms of their requirement for control genes.

Answer: We thank the referee for noticing the stringency of our criteria on the selection of control genes. The detailed criteria were listed in the section "Selection of control genes without ASD" in material and methods. The reason for such stringent selection is, we believe that, even many genes with sufficient allelic reads were not identified as ASD genes, they still may possess certain degrees of allelic biases in RNA decay rates. Genes with such potential subtle allelic biases in the control gene group might influence our further analyses. Thus, to exclude such genes and make sure all control genes are indeed without allelic biases, we applied the current stringent selection for control genes.

- For analysis of sequence features in ASD genes, the authors indicate that they selected control genes "which possessed similar density of sequence variants as ASD genes". The authors need to quantitatively define what they qualified as a similar density. They should also indicate the final number of control genes that match this criterion.

Answer: We added more details in the method section- " Specifically, based on the distribution of sequence variant density across the whole transcript in ASD genes, we randomly selected from all the control genes a subset with the same variant density distribution as ASD genes." In addition, Figure S5 (also shown below as Figure R9) shows after selection, such a subset of control genes have the same distribution of variant density as ASD genes.

Figure R9 Selection of control genes with similar density of sequence variants. The cumulative distribution function (CDF) of SNP density (number of SNPs per kb) in whole gene, 5'UTR, CDS and 3'UTR for genes with significant ASD (red) and a group of selected control genes with similar density of sequence.

Figure 3a:

- One potential confounder for the analysis presented here is that genes with more variants, and thus potentially more allelic read coverage are better powered to detect ASD. This issue would persist regardless of the minimum sufficient allelic read count requirement. The authors need to control for differences in allelic read coverage between control and ASD genes for this analysis in particular.

Answer: We thank the referees for pointing this out. We believed that the higher variant density in the ASD genes was due to the *cis*-variants causing stability divergence although this could also be due to the higher detection power from genes with high variant density, as the two referees suggested. However, as referee 3 rightfully pointed out, for the latter, what determined the detection power is allelic read counts, which could correlate with the variant density. To disentangle the two scenarios, we compared the allelic read count distribution between ASD genes and control genes. As shown in the Figure R10 below, the allelic read counts are higher in control genes than ASD genes, for both the measurement at steady state (Figure R10 A) and at all three time points (Figure R10 B). Therefore, we think the observed high variant density in the ASD genes was indeed due to the *cis*-variants.

Figure R10. Allelic read counts distribution in ASD and controls genes. Violin plot comparing the allelic read counts distribution (log2 transformed, y-axis) between ASD genes (red) and controls (blue) using the reads from steady state (A, $t=0h$) or all three time points (B, $t=0, 0.5$ and $1.5h$). Allelic read count in controls genes are significantly higher than that in ASD genes (P -value= 1.6×10^{-13} and 2.2×10^{-16} , respectively). Control genes show showing the percentage of ASD genes in those with significant ASA (blue bars) and the percentage of ASA genes in those with significant ASD (red bars) at different combinations of FDR thresholds (0.005, 0.0075, 0.01, 0.025, 0.05, 0.075 and 0.1 for ASD and ASA, respectively, two-sided Mann-Whitney U test).

Figure 3b:

- From the figure panel it appears that not all of the ASD / control genes were used. I believe this is because only genes with at least one predicted miRNA binding site were used, but this should be mentioned in the figure legend.

Answer: More details are added to the figure legend. –“Only the genes with ≥ 10 miRNA binding sites combining the two alleles together and ≥ 1 different sites between the two alleles were used.”

- In the figure legend the plots are referred to as "Barplots", this should be "Boxplots".

Answer: The error has been corrected.

- For the control genes used, were only a subset of the genes that had similar variant density used in order to match the number of ASD genes? If so, the authors need to explain how this subsetting was done and demonstrate that their result is robust to which control genes were selected.

Answer: To avoid the effect of subsetting, we used all the 1,319 control genes to compare with ASD genes. As shown in Figure R11 below, the control genes still centered around zero (P -value=0.95, two-sided Mann-Whitney U test). More details about subsetting are added to the method part, please see response above.

Figure R11. Boxplots and scatterplots showing the distribution of miRNA binding site number difference between the stable and unstable alleles for genes with significant ASD and all the controls. For controls, the difference centered around zero (P -value=0.95, two-sided Mann-Whitney U test), whereas in ASD genes, unstable alleles tend to possess more miRNA target sites than the stable alleles (P -value= 1.0×10^{-4} , two-sided Mann-Whitney U test). Only the genes with ≥ 10 miRNA binding sites combining the two alleles together and ≥ 1 different sites were used.

- The randomization of the haplotypes in the control genes to be a bit problematic, although I do understand that you can't assign one haplotype as being unstable or stable. An additional control could be to look at the difference in number of miRNA binding sites by defining haplotypes by their parental origin (BL6/SPRET). It could be for example, that one mouse strain overall has more miRNA binding sites than the other, and that this is not limited to genes where ASD is observed.

Answer: We agree with the referee that the randomization of the haplotypes is not necessary. Actually, we only performed randomization in the early version of our draft manuscript, and did not do it for our submitted manuscript, but forgot to update the method part. In other words, randomization was not performed for Fig 3B. The result with randomization can be seen from the Figure R12 A below, which shows the same trend. In addition, we also directly compared the miRNA binding site between the two strains. Figure R12 B below indicates that there is no significant difference (P -value=0.68, two-sided Mann-Whitney U test). Now the corresponding part of method section has been corrected.

Figure R12. Boxplots and scatterplots showing the distribution of miRNA binding site number difference between the stable and unstable alleles for genes with significant ASD and controls. (A) For controls, the difference centered around zero (P -value=0.69, two-sided Mann-Whitney U test), whereas in ASD genes, unstable alleles tend to possess more miRNA target sites than the stable alleles (P -value= 1.0×10^{-4} , two-sided Mann-Whitney U test). Only the genes with ≥ 10 miRNA binding sites combining the two alleles together and ≥ 1 different sites were used. (B) Boxplot and scatterplot showing the distribution of miRNA binding site number difference between B6 and SPERT for controls. The difference centered around zero (P -value=0.68, two-sided Mann-Whitney U test).

Figure 3c:

- The same concerns about selection of control genes mentioned for Figure 3b apply here.

Answer: Again, to avoid the effect of subsetting, we used all the 1,319 control genes to compare with ASD genes. As shown in Figure R13 below, the ASD genes still exhibited larger allelic differences in MFE values compared to all the control genes (P -value= 4.3×10^{-9} , two-sided Mann-Whitney U test). More details about subsetting are added to the method part.

The concern regarding randomization does not apply here since the absolute MFE difference is used.

Figure R13. Violin plots and scatterplots comparing the distribution of the absolute minimal free energy difference ($|\Delta\text{MFE}|$) between ASD genes and all the controls. The horizontal lines indicate the median. Compared to controls, ASD genes exhibited larger allelic differences (P -value= 4.3×10^{-9} , two-sided Mann-Whitney U test).

Figure 4:

- When defining the minimum fold change requirement for defining ASA significance, the manuscript text states that the same threshold as for ASD was used: "Using the same bootstrapping strategy on \log_2 fold change of allelic expression at the same threshold". It's not entirely clear to me how the $|\log_2(\text{fold change})| > 1$ threshold correlates to the $\Delta \lambda > 0.06$ threshold in terms of effect size. This is important because the level at which this cutoff is defined will directly affect the authors' conclusion that "among the 621 ASD genes, most (516 genes, 83.1%) didn't exhibit significant ASA", which is a central finding of their work. As such, I think the authors need to come up with a more robust analysis framework that is not dependent on setting arbitrary cutoffs for what constitutes significant ASA and then simply looking at gene list overlaps.

Answer: The same threshold means the same FDR. This is described more clearly in the revised version – "Using the same bootstrapping strategy on \log_2 fold change of allelic expression at the same FDR threshold".

To avoid the effect of arbitrary thresholds, we used different combinations of FDRs for ASA and ASD (0.005, 0.0075, 0.01, 0.025, 0.05, 0.075 and 0.1 for ASD and ASA, respectively). As shown in Figure R14 below, the conclusion is consistent across different thresholds. For example, when we relaxed the FDR threshold to 0.1, 706 and 3,016 genes showed significant ASD and ASA, respectively. 2,747 out of 3,016 genes with significant ASA (91.1%) didn't exhibit significant ASD. On the other hand, 437 out of 706 ASD genes (61.9%) didn't exhibit significant ASA.

This figure is added to the revised manuscript as supplementary Figure S13.

Figure R14. The role of ASD in the allelic difference of RNA abundances under different combinations of FDR thresholds. Barplots showing the percentage of ASD genes in those with significant ASA (blue bars) and the percentage of ASA genes in those with significant ASD (red bars) at different combinations of FDR thresholds (0.005, 0.0075, 0.01, 0.025, 0.05, 0.075 and 0.1 for ASD and ASA, respectively).

- It would be more informative to analyze the difference in effect size magnitude between ASA and ASD to produce a more quantitative estimate of how much ASA is observed relative to ASD. For example, the authors could quantify the amount of ASA ($|\log_2(\text{fold change})|$) observed in genes with significant ASD and compare that to control genes without ASD to get an idea of relative increase or decrease in AST affects that could potentially be compensatory to ASD effects at those genes. It would also be interesting to see if overall there is a correlation between ASD and ASA effect size.

Answer: We observed that many genes with ASD could compensate the divergence at AST and therefore led to no significant ASA. This does not mean that genes without ASD should have the same distribution of AST. Indeed, these are two different groups of genes potentially following different evolutionary trajectories of their gene expressions. However, as the referee suggested, to compare ASD genes to control genes for the distribution of ASA, we need to assume that the two groups have the similar distribution of AST. Only then, we would expect to observe lower level of ASA in genes with ASD.

Nevertheless, as the referee suggested, we compared the two groups of genes for the distribution of ASA. As shown in Figure R15 below, we did not observe significant difference between the two groups. This indicates that in general control genes have lower allelic divergence in both RNA decay and transcription.

Figure R15. Violin plots and scatterplots comparing the distribution of the gene expression fold change between ASD genes and controls. There is no significant difference between ASD genes and controls ($P\text{-value}=0.35$, two-sided Mann-Whitney U test).

- From the plot it appears that everything with $|\log_2(\text{RNABL6}/\text{RNASPRET})| > 1$ has significant ASA? Is this really the case? Surely after multiple testing correction, there would be some genes where the effect size was sufficiently large, but due to other reasons, e.g. low overall read count, that the difference is not significant.

Answer: No. There are 190 genes with $|\log_2(\text{RNABL6}/\text{RNASPRET})| > 1$ but not counted as significant ASA genes. As shown in Fig. S12B (also shown below as Figure R16), there are still many black dots (not significant genes) outside [-1, 1] range.

Figure R16. Scatterplot showing the bootstrap means (x-axis) and standard deviations (y-axis) of estimated ASE. Dashed blue lines indicate the Benjamini–Hochberg-adjusted P-value of 0.05, and dashed purple lines indicate twofold divergence of gene expression. Out of 8,815 genes (black), 1,241 (red) exhibited significant ASE.

2nd Editorial Decision

30 April 2018

Thank you for submitting your revised manuscript entitled "Pervasive allele-specific regulation on RNA decay in hybrid mice". We would be happy to publish your paper in Life Science Alliance pending final revisions to respond to the input received from reviewer #2.

As you will see, all reviewers appreciate your revised work, and reviewer #2 provides constructive input on how to finalise your submission. I would thus like to ask you to address this reviewers' comments and to provide a further revised version of your manuscript. Please let me know in case anything is unclear.

REFEREE REPORTS

Reviewer #1 (Comments to the Authors (Required)):

I am satisfied with the improvement made by the authors. I recommend its publication.

Reviewer #2 (Comments to the Authors (Required)):

I would like to thank the authors for responding to my previous comments. These responses are appropriate (with one caveat, see below). I would encourage the authors to provide their Figures R5, R6 and R7 in any published version.

Demonstrating the exponential decay of transcript abundance is important. Ultimately, Figure R4 does not do this because only the fits are shown rather than the raw data. Examples of randomly sampled decay profiles with data would provide this. Or even better format statistical tests that the data are better described by an exponential decay than a linear function.

On page 4 there is a typo that I had failed to spot previously: "Pritchard" not "Prichard".

Reviewer #3 (Comments to the Authors (Required)):

The authors' revised manuscript and rebuttal comments address the technical concerns I had with this work. I believe it is suitable for publication in its current form.

2nd Revision – authors' response

2 May 2018

Reviewer #2:

I would like to thank the authors for responding to my previous comments. These responses are appropriate (with one caveat, see below). I would encourage the authors to provide their Figures R5, R6 and R7 in any published version.

Answer: We thank the referee for the suggestions. Figure R5 and R6 are added to the revised manuscript as supplementary Figures S7 and S10, respectively. Figure R7 can serve as a quality control of our data. However, it is not the focus of this study, therefore we do not feel like to include it in the revised manuscript.

Demonstrating the exponential decay of transcript abundance is important. Ultimately, Figure R4 does not do this because only the fits are shown rather than the raw data. Examples of randomly sampled decay profiles with data would provide this. Or even better format statistical tests that the data are better described by an exponential decay than a linear function.

Answer: It has been accepted that RNA decay follows a first-order kinetics (Perez-Ortin et al., 2013; Ross, 1995), i.e. the decay rates depend on the cellular concentrations of RNA molecules. Therefore, RNA decay fits with an exponential model,

$$\ln(B_t) = B_0 - kt$$

Where B_t is the RNA abundance at time t , B_0 is the RNA abundance at 0 hr, k is the gene-specific RNA decay rate constant.

In the Figure RII-1 below, left and right part shows the raw data and exponential fits for each gene, respectively. Here, for this analysis, we ignored the sequence variants between the two strains and mapped the F1 hybrid reads to the standard mouse genome (B6) using Tophat2. Then Cufflinks was used to estimate the FPKM value for each gene in each sample from different time points. For each gene, the mean value of the two replicates was used. 3949 expressed genes (FPKM>1 at 0h) with degradation signals (FPKM at 0.5h < FPKM at 0h and FPKM at 1.5h < FPKM at 0h) were retained for this analysis.

Figure RII-1. Profiles of decay rates. Distribution of genome-wide decay profiles across the timecourse experiment (x-axis), where each decay curve shows the decrease in gene expression level (y-axis, raw data and exponential fits for left and right figure, respectively) relative to the steady state. Each line represents the gene-specific decay profile.

As the reviewer suggested, to compare whether the linear model or exponential model fits out data better, we fitted our data using the two models separately and estimated the Akaike information criterion (AIC) for the two models (for AIC, please see <http://stat.ethz.ch/R-manual/R-devel/library/stats/html/AIC.html>). As shown in Figure RII-2, for the majority of genes, exponential model outperforms linear model.

Figure RII-2. Comparison of AIC based on linear model (y-axis) to that based on exponential model (x-axis).

On page 4 there is a typo that I had failed to spot previously: "Pritchard" not "Prichard".

Answer: The change has been made in the new version.

Reference

Perez-Ortin, J.E., Alepuz, P., Chavez, S., and Choder, M. (2013). Eukaryotic mRNA decay: methodologies, pathways, and links to other stages of gene expression. *J Mol Biol* 425, 3750-3775.

Ross, J. (1995). mRNA stability in mammalian cells. *Microbiol Rev* 59, 423-450.

Accept

3 May 2018

Thank you for submitting your Research Article entitled "Pervasive allele-specific regulation on RNA decay in hybrid mice".

I appreciate the response you provided to reviewer #2's remaining concern, and it is a pleasure to let you know that your manuscript is now accepted for publication in Life Science Alliance.

Congratulations on this interesting work.